# Anti-Hyperglycemic Medication Management in the Perioperative Setting: A Review and Illustrative Case of an Adverse Effect of GLP-1 Receptor Agonist

**DOI:** 10.3390/jcm13206259

**Published:** 2024-10-20

**Authors:** Abby R. Goron, Courtney Connolly, Arielle N. Valdez-Sinon, Ashley Hesson, Christine Helou, Gregory W. Kirschen

**Affiliations:** 1Department of Obstetrics and Gynecology, Vanderbilt University Medical Center, Nashville, TN 37232, USA; abby.goron@vumc.org; 2Department of Gynecology and Obstetrics, Johns Hopkins Hospital, Baltimore, MD 21287, USA; cconno23@jhmi.edu (C.C.); avaldez2@jhmi.edu (A.N.V.-S.); 3Department of Obstetrics and Gynecology, University of Michigan, Ann Arbor, MI 48109, USA; ahesson@med.umich.edu; 4Department of Obstetrics and Gynecology, Greater Baltimore Medical Center, Towson, MD 21204, USA; christine.m.helou@gmail.com; 5Division of Maternal-Fetal Medicine, Department of Obstetrics and Gynecology, University of Pennsylvania, Philadelphia, PA 19104, USA

**Keywords:** pharmacovigilance, anti-hyperglycemic agents, GLP-1 receptor agonists, perioperative care

## Abstract

A host of anti-hyperglycemic agents are currently available and widely prescribed for diabetes and weight loss management. In patients undergoing surgery, use of these agents poses a clinical challenge to surgeons, anesthesiologists, and other perioperative care providers with regard to optimal timing of discontinuation and resumption of use, as well as possible effects of these agents on physiology and risk of postoperative complications. Here, we provide a comprehensive review of anti-hyperglycemic medications’ effects on physiology, risks/benefits, and best practice management in the perioperative setting. Additionally, we report an illustrative case of small bowel obstruction in a patient taking semaglutide for 6 months prior to an otherwise uncomplicated laparoscopic hysterectomy and bilateral salpingo-oophorectomy. This review is meant to serve not as a replacement of, but rather as a consolidated complement to, various society guidelines regarding perioperative anti-hyperglycemic agent management.

## 1. Introduction

The number of patients with diabetes mellitus undergoing surgery worldwide is increasing. One review from 2004 estimated that 15–20% of surgical patients are diabetic [1], and these numbers have likely increased over the past two decades. Physiologic stress from surgery induces insulin resistance and increased glucose production, resulting in stress hyperglycemia [2]. Persistently elevated glucose levels may promote immune dysfunction, susceptibility to infections, poor wound healing, endothelial dysfunction, and thrombosis [3]. Alternatively, persistent perioperative hypoglycemia also has several significant consequences, including cardiac arrhythmias and electrolyte disturbances. Patients at higher risk of perioperative hypoglycemia, in particular, include those with type 1 diabetes, longer duration of diabetes, history of hypoglycemia, poor nutritional status, low body mass index (BMI), and patients who use insulin, sulfonylureas, or a meglitinide [4,5]. Thus, perioperative glycemic control is paramount.

Optimal perioperative glucose targets have not been firmly established, however the American Diabetes Association (ADA) has endorsed a target glucose range of 80 to 180 mg/dL, with a target of 140 to 180 mg/dL in critically ill patients [6,7]. Ultimately, while several strategies exist to maintain a patient’s blood glucose level in the goal range, there is no one clearly superior approach [5,8]. The decision of perioperative management will be influenced largely by which antihyperglycemic agent(s) patients are utilizing in the outpatient setting, as well as other patient-specific factors of the individual patient, their provider, and the clinical setting.

In this review article, we systematically examine the effects of various anti-hyperglycemic medications on physiology among surgical patients. The goal of this review is to highlight perioperative considerations for each of the anti-hyperglycemic agents currently employed in practice to serve as a primer for surgeons, anesthesiologists, and other perioperative care team members. We present this topic from the vantage point of obstetrician–gynecologists, and acknowledge that other perioperative team members, such as general surgeons, internists, anesthesiologists, may have differing views and opinions, all of which fall within the standard of care.

## 2. Anti-Hyperglycemic Agents

### 2.1. Insulin

Insulin is a hormone produced by pancreatic β-cells that promotes cellular glucose uptake. Exogenously administered insulin is a critical tool in the management of diabetes—in 2021, the Centers for Disease Control and Prevention (CDC) reported that 12.3% of all United States adults with diabetes started using insulin within a year of their diagnosis [9]. Thus, providers should be aware of optimal management of insulin in the perioperative setting.

Insulin can be administered subcutaneously or via a continuous pump. Injectable insulin is commonly administered as one or several basal doses, prandial doses, and sometimes correctional doses based on blood sugar target levels or carbohydrates consumed. Patients with type 1 diabetes lack endogenous insulin; they are at risk of ketoacidosis if sufficient insulin is not supplied. For these patients, even perioperatively with minimal food intake, basal insulin must be administered [10]. For patients who only take once-daily basal insulin (i.e., neutral protamine Hagedorn (NPH), glargine), this may often be continued as normal without any change in dosing if the patient has demonstrated stable outpatient blood glucose levels in a safe range. For patients with a history of hypoglycemia or low-normal glucose levels, it is often recommended to reduce the dose by 10–25% the night prior to surgery, or morning of surgery if dosing is in the morning [11]. Patients taking twice daily dosing of basal insulin can follow the same above principles and decrease both their morning and evening dose of insulin by 10–25%. For patients who administer high doses of basal insulin (total daily dose exceeding 80 units; >60% of total daily dose) or are at risk of hypoglycemia, consider reducing the dose of basal insulin by 50–75% to minimize risk of hypoglycemia [11].

In the case of intermediate-acting insulin such as NPH, the usual dose is administered the evening before surgery, with 50% of the dose administered the morning of surgery. Patients using premixed insulin (i.e., NPH/regular 70/30) should receive long-acting insulin the evening before surgery and not their premixed formulation. If this is not possible, the premixed solution should be dose-reduced by 20% the evening prior to surgery and 50% on the morning of surgery, or held completely if morning blood glucose is <120 mg/dL [12]. For patients who use prandial insulin, this should be held while fasting.

The proportion of patients using insulin pumps is also increasing. Studies on their use perioperatively are largely retrospective analyses; however, they demonstrate no clear differences in glucose control between those using an insulin pump versus intravenous (IV) insulin [13]. Continuing insulin pump use intra-operatively should be limited to procedures lasting less than 2 h. Patients undergoing more complex procedures should be transitioned to a basal–bolus regimen [14].

Intra-operatively, insulin-dependent patients can receive insulin subcutaneously dosed every few hours as needed. Blood glucose should be checked every hour, or more often if the value is less than 100 mg/dL. General guidelines are that insulin should be initiated once blood glucose level is above 180 mg/dL [15]. Additionally, dextrose-containing IV fluid should be given intra-operatively to avoid metabolic starvation [16]. For longer cases or more medically complex patients, IV insulin infusion may be required, either as separate glucose and insulin infusions or in a combination glucose–insulin–potassium infusion [16]. Studies comparing subcutaneous versus IV infusion of insulin have noted more variability in blood glucose with subcutaneous routes [17]. IV insulin has been demonstrated to be safe and readily titratable due to short half-life of 5 to 10 min (compared to half-life of aspart insulin of ~80 min). To administer IV insulin safely, blood glucose should be checked every one to two hours, and electrolytes (potassium and bicarbonate) should be monitored. In cases of hypoglycemia, insulin infusions may be paused. However, in the setting of insulin-dependent patients, glucose can be administered along with insulin to prevent ketosis [14].

Postoperatively, blood glucose should be monitored every two to three hours until the patient is awake and alert. Insulin infusions should be continued in postoperative patients who do not resume eating immediately. Once a patient is tolerating solid food, they can be transitioned to subcutaneous insulin, with the first subcutaneous dose given before discontinuation of the insulin infusion. For patients treated only with subcutaneous insulin intra-operatively, subcutaneous insulin and IV dextrose should be continued. Once sufficient oral intake can be tolerated, a basal–bolus regimen can be restarted. Ambulatory patients may be discharged with instructions to resume their usual home insulin regimen. For non-critically ill inpatients requiring hospital admission, the preferred approach for insulin therapy is a basal regimen with correctional insulin, typically given when glucose levels are >150 mg/dL, although this may vary [18].

For those tolerating oral intake, insulin regimens should include basal, nutritional, and correctional insulin. Insulin dosages postoperatively can be adjusted based on either weight or regimen, with several suggested algorithms. In an average-weight patient, the total daily dose (TDD) of insulin is 0.4 to 0.5 units/kg/day. Two thirds of the TDD will be administered as basal, and one third will be administered as nutritional insulin with three meals. Blood glucose levels are typically monitored before meals and at bedtime, and correctional insulin administered accordingly [12,19].

### 2.2. Insulin Sensitizers: Biguanides and Thiazolidinediones

Metformin is a biguanide medication that functions primarily by decreasing hepatic glucose output and inhibiting gluconeogenesis [20]. It is the most widely used glucose-lowering agent and one of the most prescribed drugs worldwide [21]. Providers should be aware of the perioperative management of metformin given its widespread use. A commonly cited concern regarding use of metformin perioperatively is the potential for lactic acidosis. Perioperative administration should be deferred in patients with impaired kidney function (glomerular filtration rate < 30 mL/min), heart failure, or chronic liver disease, given risk of accumulation of plasma metformin levels, leading to reduced lactate clearance [22]. However, randomized control trials examining metformin use in the perioperative period did not demonstrate significantly affected lactate levels in coronary artery bypass or noncardiac surgery patients with type 2 diabetes [23,24]. The risk of metformin-associated lactic acidosis is thought to be extremely low (less than 10 cases per 100,000 patient-years) [22].

Research has also demonstrated that surgical patients utilizing metformin have unchanged, if not improved, outcomes, including fewer incidences of myocardial infarction, stroke, and other diabetes-related endpoints, some of which demonstrated persistent risk reduction for up to 10 years [22,25]. Several reports of safe postoperative metformin use have also been published in the post-cardiac surgery population [23,26]. It is critical to avoid unnecessary cessation of metformin perioperatively, to allow patients the benefits demonstrated [27]. Adverse effects may include nausea, vomiting, diarrhea, and abdominal pain [28].

Ultimately, for same-day minor surgery, it is recommended to continue metformin, except in patients with renal dysfunction or in interventions requiring significant contrast administration, use of nonsteroidal anti-inflammatory drugs (NSAIDs), angiotensin-converting enzyme inhibitors, or angiotensin II receptor blockers [29,30]. For major surgeries, metformin should be stopped the day prior to surgery. Patients with a preoperative hemoglobin A1c below 7% should have blood glucose monitored every two hours intraoperatively. If a patient develops hyperglycemia, short- or rapid-acting insulin may be administered based on glucose levels. Once a patient has resumed oral intake and renal function has stabilized, metformin may be restarted [31].

The thiazolidinedione drug class acts by increasing insulin sensitivity, likely by binding and activating predominantly the gamma isoform of peroxisome proliferator-activated receptors (PPAR-γ), which, therefore, affects gene transcription important in metabolism [32]. The PPAR-γ isoform is present mainly in adipose tissue, pancreatic β-cells, the central nervous system, and the vascular endothelium, and the insulin-sensitizing effect is likely due to the effect in adipose cells [33]. There is also a PPAR-α isoform localized to the liver, heart, skeletal muscle, and vasculature, and glitazones with activation of the PPAR-α isoform have been linked to adverse cardiovascular outcomes, including ischemic heart disease [34,35]. Pioglitazone is the only currently available agent of this class approved for use in the United States.

Due to the high risk of different adverse effects, including heart failure, fluid retention, weight gain, fractures, and possible increased risk of bladder cancer, glitazones are not commonly prescribed and are reserved for second- or third-line medication treatment [36,37,38]. However, one benefit of glitazones is a decreased incidence of hypoglycemia when compared to other anti-hyperglycemic agents [39]. Therefore, it is generally accepted that glitazones can be continued through the perioperative period and taken on the day of surgery [31,40]. There is currently no evidence in the literature of postoperative complications specifically associated with the use of thiazolidinediones.

### 2.3. Secretagogues: Sulfonylureas and Meglitinides

Both sulfonylureas and meglitinides serve as insulin secretagogues, increasing secretion of insulin from pancreatic β-cells to lower glucose levels. Sulfonylureas, as a class, work on the sulfonylurea receptor, which is an adenosine-triphosphate-sensitive potassium channel in pancreatic β-cells that controls the release of insulin from these cells [41]. They may be chosen as initial therapy for patients with contraindications to metformin and maturity-onset diabetes of the young or monogenic diabetes [42]. Due to their mechanisms of action, there is a high risk of hypoglycemia, with a rate of hypoglycemia 2.4 times higher in patients receiving a sulfonylurea versus other anti-hyperglycemic agents in one study [43]. Other adverse events include modest weight gain, nausea, skin reactions, and elevated liver function tests [44].

Sulfonylureas can be stratified by duration of action, with shorter-acting medications placing patients at lower risk for hypoglycemia. Shorter-lasting sulfonylureas include glipizide, whereas glyburide is a longer-lasting sulfonylurea. Due to the varying half-lives of different sulfonylureas, management can be individualized at the patient level based on fasting duration, however it is generally accepted to hold shorter-lasting sulfonylureas the night before or morning of surgery, and longer-lasting sulfonylureas for 48 to 72 h, resuming postoperatively when diet is resumed [31].

Meglitinides work similarly via a different adenosine-triphosphate-sensitive potassium channel in pancreatic β-cells [45]. Common meglitinides include repaglinide and nateglinide. Due to their rapid onset of action and shorter duration, they are recommended to be administered with meals. For example, the half-life of repaglinide is approximately 1 h [46,47]. Meglitinides may be chosen, like with sulfonylureas, when there is a contraindication to use of metformin or as an add-on therapy to metformin. Adverse effects include a similar degree of weight gain as sulfonylureas; however, there is a lower risk of hypoglycemia [47]. Due to the short half-life of meglitinides, it is recommended that they be held the morning of surgery and restarted postoperatively when diet is resumed [31].

### 2.4. Alpha Glucosidase Inhibitors

Alpha glucosidase inhibitors work by decreasing absorption of glucose in the upper intestine by inhibiting the enzymes that convert complex polysaccharides into monosaccharides. This also allows more complex polysaccharides to transit to the distal intestine, which increases secretion of glucagon-like peptide 1 (GLP-1) [48]. Therefore, their mechanism of action is directly linked to ingestion of carbohydrates and works well for patients with postprandial hyperglycemia. Adverse effects include flatulence and elevated liver function tests [49,50]. Alpha glucosidase inhibitors available in the United States include acarbose and miglitol. Due to their mechanism of action, alpha glucosidase inhibitors should be held when fasting and can be resumed postoperatively when a diet is resumed [40,51].

### 2.5. Peptide Analogs: GLP-1 Receptor Agonists and DPP-4 Inhibitors

#### 2.5.1. GLP-1 Receptor Agonists

As the medical landscape continues to evolve with the introduction of novel medications, gynecologists must stay abreast of pharmacological complexities that may affect their surgical patients. Of particular note, glucagon-like peptide 1 receptor agonists (GLP-1RAs) have revolutionized type 2 diabetes mellitus care and medical weight loss treatment, though their effects on physiology in patients undergoing surgery remain incompletely understood [52,53].

GLP-1 is a compound produced by the small intestine after meal ingestion. It binds to a GLP-1 receptor, which is expressed in various tissues, the most important of which are the pancreatic β-cells (leading to production of insulin) and the gastric mucosa (leading to inhibition of gastric emptying and appetite suppression) [54]. Endogenous GLP-1 has a short half-life of 1–2 min due to N-terminal degradation by dipeptidyl peptidase-4 (DPP-4), however synthetic GLP-1Ras are variably resistant to DPP-4 degradation, extending their half-lives to 13 h (liraglutide) to 7 days (semaglutide) [55,56].

GLP-1RAs, though effective in managing diabetes and promoting weight loss, carry notable gastrointestinal side effects, including nausea, vomiting, and diarrhea. More concerning are the rare but severe complications, such as pancreatitis, bowel obstruction, and gastroparesis. Two recent studies have hinted at an increased risk of bowel obstruction in those taking GLP-1RAs but did not provide temporal resolution with regard to pharmacokinetics in relation to timing of surgery [57,58]. Here, we underscore the importance of vigilance, particularly in the abdominal surgical setting, and highlight a recent case underscoring the potential adverse effects of GLP-1Ras in surgical patients, prompting a discussion on prevention and management strategies. The patient’s written informed consent was obtained for this case report.

The case involves a 52-year-old female with a history of type 2 diabetes mellitus undergoing total laparoscopic hysterectomy with bilateral salpingo-oophorectomy for abnormal uterine bleeding. The patient had been treated with semaglutide for six months leading up to the surgery, with the last dose received 10 days prior to surgery. Intraoperatively, adhesions of the small bowel and colon to the anterior abdominal wall in the left upper quadrant were present, however the pelvis was easily visualized and surgery completed without any adhesiolysis. Despite minimal intraoperative bowel manipulation, she developed a delayed small bowel obstruction (SBO) postoperatively, necessitating prolonged hospitalization. Upon her re-presentation on postoperative day 12, abdominal computed tomography (CT) scan with IV contrast demonstrated a mechanical SBO with a transition point in the anterior central abdomen, with dilated proximal small bowel up to 3.5 cm and decompressed small bowel distally (Figure 1A).

She was managed conservatively with nasogastric tube (NGT), and after continued lack of flatus on hospital day 3, she underwent abdominal X-ray with an oral contrast challenge, which demonstrated contrast in the ileum after 14 h (Figure 1B). In this setting, the NGT was removed. On hospital day 5, she had a bowel movement and was advanced to a regular diet. However, for several days afterwards she continued to report intermittent nausea, persistent abdominal pain, and intermittent return of flatus. She underwent a second CT abdomen/pelvis with IV and oral contrast on hospital day 11, demonstrating imaging resolution of SBO and notation of mild to moderate stool burden (Figure 1C).

Given lack of mechanical obstruction on imaging but persistent symptoms, Gastroenterology was consulted, and suspected that previous semaglutide use may have contributed to a dysmotility picture. Additional contributing factors were proposed, such as the patient’s opioid use for postoperative pain (although her use was limited during hospitalization, and these agents are known to contribute more to ileus than obstruction), as well as prolonged hospitalization and relative decreased mobility during her stay. Confounding factors for development of obstruction also include the patient’s abdominal surgical history, notable for an open small bowel resection, laparoscopic cholecystectomy, and bilateral tubal ligation. Gastroenterology recommended a bowel regimen as well as a pro-motility agent. As the patient was allergic to metoclopramide, she was treated with naloxegol, a mu-opioid receptor antagonist, and was discharged home on hospital day 11 in stable condition [59]. On follow-up several weeks later, she had stopped the naloxegol and was doing well with consistent bowel function aided by polyethylene glycol.

This case highlights the need for heightened awareness among clinicians regarding the potential risks associated with GLP-1RA use in surgical patients. While limited data exist, there is a plausible link between these medications and gastrointestinal complications, particularly in individuals with a history of bowel surgery. In this instance, the patient’s prior small bowel resection likely predisposed her to obstruction, potentially exacerbated by the use of GLP-1RA given its well-established pattern of slowed gastric emptying and motility. This case corroborates slowed gastrointestinal transit (oro-ileal time of 14 h) in a postoperative patient taking semaglutide, in contrast to previous studies of other bioactive peptides showing normal oro-ileal time of 84 min (mean) to 5 h (maximum) in healthy volunteers [60,61].

Preoperative management guidelines, such as those outlined by the American Society of Anesthesiologists (ASA), recommend holding GLP-1RAs before surgery (for 1 week prior if weekly dosed, and for 1 day prior if daily dosed) and adopting full stomach precautions due to the medications’ effects on gastric emptying [62]. The ASA guidelines do not suggest an optimal duration of fasting for this patient population or modification of standard recommendations related to oral intake prior to surgery. If the medication was not held, they recommend consideration of gastric ultrasound and even delaying the procedure if the stomach appears full on ultrasound, or if significant gastrointestinal symptoms are present.

One of our institution’s preoperative management policies for patients taking GLP-1RAs includes recommendation of a clear liquid diet for 12 h before the usual nil per os (NPO) time (for example noon the day before surgery if NPO would start at midnight), rather than typical NPO at midnight or allowance of clear liquids up to 2 h prior to surgery in patients without contraindications, as is generally used in accordance with ASA guidelines and enhanced recovery after surgery (ERAS) protocols. Our guidelines also recommend avoiding ERAS preoperative carbohydrate beverages and considering delaying the procedure if the patient is experiencing gastrointestinal symptoms. However, our case suggests that even with adherence to these guidelines, patients may still experience postoperative gastrointestinal complications; our patient had held semaglutide for 10 days, i.e., 1.5 half-lives of the drug.

Moreover, it is important to highlight that gastrointestinal complications may have a delayed presentation, as demonstrated in our case. A recent large retrospective observational cohort study comparing GLP-1RA users to non-users (those who used a non-GLP-1RA oral anti-hyperglycemic agent) for 6 months prior to surgery did not find an increased risk of ileus development within 7 days of surgery [63]. However, that study did not examine longer-term risk for delayed gastric emptying, which will be crucial in understanding the overall post-operative recovery for this population. 

Moving forward, it is crucial for surgeons and anesthesiologists to reevaluate preoperative protocols for patients treated with GLP-1RA, including consideration of extended washout periods or delaying surgery if necessary. Additionally, early initiation of pro-motility agents, such as metoclopramide or naloxegol (in patients taking concomitant opioids), may mitigate the bowel-slowing effects of GLP-1RAs in the immediate or delayed postoperative period, although this is an off-label indication.

#### 2.5.2. DPP-4 Inhibitors

Dipeptidyl peptidase-4 (DPP-4) inhibitors prevent the degradation of endogenous GLP-1, leading to an increased activity of GLP-1 to stimulate insulin secretion and suppress glucagon secretion by the pancreas [31]. This drug class includes oral agents, such as sitagliptin, saxagliptin, linagliptin, and alogliptin. A benefit of the use of DPP-4 inhibitors over other diabetic medications is that it has a low risk of hypoglycemia (unless a sulfonylurea is being co-administered) [64]. DPP-4 inhibitors are typically well tolerated and considered to have an overall good safety profile. Rare reported side effects include pancreatitis, arthralgia, bullous pemphigoid, heart failure [64]. More commonly reported side effects may include headache, nasopharyngitis, and upper respiratory tract infections [65]. Unlike other anti-hyperglycemic agents, DPP-4 inhibitors do not need to be held preoperatively [29,31,40].

The DPP-4 inhibitor drug class has a heterogenous pharmacokinetics profile [66]. Unlike the other DPP-4 inhibitors, saxagliptin is primarily metabolized by a single cytochrome P450 isomer (CYP3A4/A5). Thus, administration of any strong inducer or inhibitor of this metabolic enzyme can significantly impact the pharmacokinetics of saxagliptin [67]. Overall, DPP-4 inhibitors do not have clinically significant drug–drug interactions and can be used with other anti-hyperglycemic agents, including metformin and thiazolidinediones, in addition to other medications common for diabetic patients with comorbidities (i.e., antihypertensives and statins) [67].

During coronary artery bypass graft surgery, patients with diabetes initiated on sitagliptin the day prior to surgery had similar intraoperative blood glucose concentrations compared to controls [68]. In the same study, there was no difference in rates of postoperative hyperglycemia, length of stay, complications, readmission, or surgical reinterventions [68]. Another study involving diabetic patients undergoing coronary artery bypass grafting demonstrated postoperative DPP-4 inhibitor administration reduced the incidence of major adverse cardiac and cerebrovascular events compared to the control group [69]. There has not been any other evidence that DPP-4 inhibitors reduce morbidity or mortality in the perioperative setting. However, there is strong evidence from randomized control trials that DPP-4 inhibitors can be used for inpatient management of blood glucose concentrations. Use of linagliptin in the postoperative setting decreases blood glucose concentration similarly to basal–bolus insulin regimens, with reduction of hypoglycemic events (1.6% vs. 11%) and decreased supplemental insulin injections compared to a traditional basal–bolus insulin regimen [70]. Similar results have been found in non-surgical hospitalized diabetic patients with administration of saxagliptin [71]. Hospitalized diabetic patients taking saxagliptin also had lower glycemic variability compared to the use of basal–bolus insulin therapy [71].

There is currently no evidence in the literature of postoperative complications specifically associated with the use of DPP-4 inhibitors. As these drugs do not have the same impacts on gastrointestinal motility as other anti-hyperglycemic agents, like GLP-1RAs, the risk of perioperative aspiration is not increased with use of DPP-4 inhibitors. Thus, there is no evidence to suggest that DPP-4 inhibitors need to be held in the perioperative setting. Continued research is needed to fully elucidate perioperative concerns or complications related to DPP-4 inhibitor use. The literature available currently primarily applies to patients with diabetes undergoing cardiac procedures without any studies specifically investigating patients undergoing gynecologic surgeries. Based upon current literature and recommendations available, it can be concluded that DPP-4 inhibitors do not need to be held prior to surgery and can be used for inpatient management of blood glucose concentration following surgery. Though they may not have benefits for reducing postoperative complications or length of stay, DPP-4 inhibitors have the benefits of reducing the amount of correctional insulin needed and are associated with less hypoglycemic events in the postoperative setting.

### 2.6. SGLT2 Inhibitors

Sodium glucose cotransporter-2 inhibitors (SGLT2is) act in the proximal renal tubule, preventing resorption of glucose [72]. Sodium glucose cotransporter-2 channels (SGLTs) represent one of the six named sodium ion co-transporter proteins that modulate the filtration of a variety of substances, including glucose [73]. By increasing the excretion of glucose, SGLT2is directly counter hyperglycemia. They also have a modest antihypertensive effect, thought to be largely attributable to the diuresis induced via glucosuria. SGLT2is are additionally associated with a variety of extrarenal effects; for example, SGLT2is have a protective effect on the β-cells of the pancreas, contributing to their beneficial profile in the setting of type 2 diabetes [74]. They have been noted to be cardioprotective as well. In individuals with and without diabetes, SGLT2i use decreases the risk of heart failure progression and lessens the risk of cardiovascular morbidity/mortality [75,76].

The mechanisms of SGLT2is’ extrarenal effects are poorly understood. Though SGLTs are found outside of the kidney, their distribution does not fully explain the cardiac manifestations of these medications in particular, as SGLT-2s are not found in high concentrations in myocardium [77,78]. SGLT2is are currently Food and Drug Administration (FDA)-approved for treatment of type 2 diabetes, as part of guideline-directed medical therapy in heart failure, and to reduce complications in chronic kidney disease. More indications are likely in the future, as there are ongoing investigations of SGLT2is to treat an array of chronic conditions with secondary cardiovascular implications.

The pharmacokinetics of specific SGLT2is vary. Though they all have maximum absorption times on the order of 1–2 h, their bioavailability values range from 68% (canagliflozin) to 100% (ertugliflozin) [79]. Half-lives for the most common SGLT2is (dapagliflozin, canagliflozin, and empagliflozin) are all approximately 12 h. Metabolism is hepatic via uridine 5′-diphosphate-glucuronosyltransferases, with individual variation mediated, in part, by genetic and epigenetic factors [79]. SGLT2is are excreted in similar proportions in urine and feces [80]. Common adverse effects include urinary and genital tract infections potentiated by the glucose-rich urine generated by SGLT-2 inhibition; euglycemic diabetic ketoacidosis (eDKA) is a serious, well-documented adverse event, but it is relatively rare in type 2 diabetes (SGLT2is are not indicated for the treatment of type 1 diabetes) [81,82].

The FDA recommends discontinuation of SGLT2is 72 h prior to surgery, largely due to the concern for eDKA [83]. In the perioperative context specifically, eDKA is thought to occur frequently in those who did not preoperatively hold their SGLT2i [84]. It is unclear whether this risk is exclusively borne by those with type 2 diabetes or if all patients taking SGLT2is can develop eDKA perioperatively, though there is some evidence suggesting that patients with diabetes represent the majority of eDKA cases [83,85]. Estimates of the frequency of eDKA vary widely, even within the diabetic population, with a multicenter series of 759 patients reporting a 0% rate of eDKA in perioperative patients [85] compared to a single-center series of 463 patients showing evidence of ketoacidosis in all patients who did not hold an SGLT2i. Differences in ascertainment of eDKA and means of excluding other perioperative causes for acidosis are the likely causes of these discrepant estimates. Guidance for anesthesiologists emphasizes having a high index of suspicion for eDKA and reaffirms the FDA recommendation to discontinue SGLT2is several days prior to anticipated surgery [31]. These guidelines also highlight the potential for dehydration in patients taking SGLT2is on the day of surgery.

As the indications for SGLT2is continue to expand, further studies on optimal perioperative management are needed to better define the risks of surgery with and without perioperatively holding SGLT2is.

## 3. Conclusions

This review is meant to serve as a reference for surgeons, anesthesiologists, and other perioperative care providers managing surgical patients on various anti-hyperglycemic agents. Regardless of which anti-hyperglycemic agent a patient is using, patients on these agents and with diabetes mellitus should ideally be scheduled as the first surgical case of the day, to reduce time needed to be NPO. By enhancing awareness and implementing tailored management strategies, we can mitigate the risks of these medications and optimize postoperative recovery. Table 1 summarizes surgery-relevant facts regarding various anti-hyperglycemic medications.

## Figures and Tables

**Figure 1 jcm-13-06259-f001:**
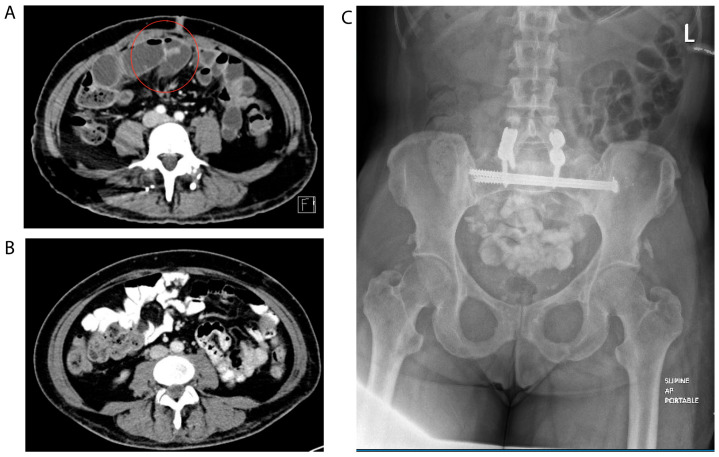
(**A**). Initial CT of the abdomen and pelvis upon readmission demonstrating small bowel obstruction. Imaging findings of mechanical small bowel obstruction with transition point anterior central abdomen (red circle). Immediately proximal to this location, there is some fecal content within the dilated small bowel loop up to 3.5 cm. Beyond this area, the small bowel is under-distended. (**B**). Final computed tomography of abdomen pelvis prior to discharge demonstrating interval resolution of small bowel obstruction. Interval resolution of previously observed small bowel obstruction with decompression of small bowel loops. No free air. Air-fluid level in the stomach may relate to gastric dysmotility. (**C**). Small bowel follow through on hospital day 2 demonstrating evidence of resolving partial small bowel obstruction. Oral contrast in nondilated ileum. Small amount of nonspecific gas in jejunum and left aspect of the colon. Evidence suggestive of resolving partial small bowel obstruction.

**Table 1 jcm-13-06259-t001:** Perioperative considerations for the various anti-hyperglycemic agents.

Class	Mechanism of Action	Generic Names	Half-Life	Hypoglycemia Risk	Before Surgery	After Surgery
Insulin	Promotes cellular uptake of glucose	Rapid acting: aspart, lispro Short acting: regular insulin Intermediate acting: isophane insulin Long-acting: detemir, degludec, glargine	1–1.5 h 5–7 min 4.5 h 5–25 h	High risk	Dose-reduce basal insulin by 10–25% (exact amount varies) Hold prandial insulin	Resume basal-bolus regimen when tolerating PO
Biguanides	Decreases hepatic glucose output Inhibits gluconeogenesis	Metformin Metformin XR	6–18 h 24 h	Low risk	Hold 1 day prior to surgery if major surgery May continue if minor surgery	Hold if inpatient and impaired renal function
Thiazolidinediones	Enhances insulin sensitivity by activating PPAR-γ	Pioglitazone Rosiglitazone	3–7 h 3–4 h	Low risk	Continue perioperatively	Continue perioperatively
Sulfonylureas	Stimulates insulin secretion by inhibiting K-ATP channel, promoting calcium influx into β-cells	Chlorpropamide Glimepiride Glipizide Glyburide	36 h 5–8 h 2–7 h 10 h	High risk	Hold long-acting 48–72 h preoperatively; hold short-acting the night before or day of	Resume when tolerating PO
Meglitinides	Stimulates insulin secretion by inhibiting K-ATP channel, promoting calcium influx into β-cells	Nateglinide Repaglinide	1 h	Moderate risk	Hold the morning of surgery	Resume when tolerating PO
Alpha glucosidase inhibitors	Inhibits intestinal absorption	Acarbose Miglitol Voglibose	3–9 h 30 min to 2 h 4 h	Low risk	Hold the morning of surgery if fasting	Resume when tolerating PO
GLP-1 receptor agonists	Increase glucose-dependent insulin secretion Slow gastric emptying Reduce postprandial glucagon and food intake	Dulaglutide Exenatide Liraglutide Lixisenatide Semaglutide	80 h 1.5–4 h 11–15 h 3 h 165 h	Low risk	Hold for 1 week if weekly dosing, 1 day if daily dosing	May resume when tolerating PO
DPP-4 inhibitors	Block the action of DPP-4 enzyme, which then increases GLP-1 and GIP levels	Alogliptin Linagliptin Saxagliptin Sitagliptin	21 h 12 h 2.5 h 12 h	Low risk	May continue to use preoperatively	May use inpatient to reduce insulin requirements
SGLT-2 inhibitors	Inhibits renal glucose reabsorption	Bexagliflozin Canagliflozin Dapagliflozin Empagliflozin Ertugliflozin Sotagliflozin	12 h 10–13 h 13 h 12 h 16 h 5–10 h	Low risk	Discontinue 72 h prior to surgery	Resume when tolerating PO

Abbreviations: DPP-4: dipeptidyl peptidase-4; GIP: glucose-dependent insulinotropic polypeptide; GLP-1: glucagon-like peptide-1; K-ATP: potassium adenosine triphosphate; PPAR-γ: peroxisome proliferator-activated receptor gamma; PO: per os; SGLT-2: sodium-glucose transporter-2.

## Data Availability

Data from the case report are available upon reasonable request.

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
