# Peer review of "Anti-Hyperglycemic Medication Management in the Perioperative Setting: A Review and Illustrative Case of an Adverse Effect of GLP-1 Receptor Agonist"

_jcm, 2024, doi:10.3390/jcm13206259_

Round 1

Reviewer 1 Report

Comments and Suggestions for Authors

Comments to authors

The authors have written a hybrid article between a narrative review and a case report. While it may be of interest, I believe the manuscript needs improvement.

·        First, the paragraphs are too long. Paragraphs should generally be no longer than 12-15 lines. In addition, each paragraph should focus on one topic (or several closely related topics). In other words, the manuscript should be restructured by improving the structure, length, and coherence of the paragraphs.

·        Second, Table 1 should be placed at the beginning of Section 2 of the manuscript, not after the conclusions. It is the guideline proposed by the authors and should be mentioned at the beginning of Section 2 so that readers can quickly visualise the information in the entire manuscript.

·        Section 2 "Insulin", Section 2.1 "Metformin", etc. This structure is confusing. It implies that metformin is a type of insulin. I have to assume that section 2 is about the most commonly used hypoglycaemic drugs. If so, you can call it "Major hypoglycaemic drugs". If the authors mean something else, call it something else. But don't call section 2 "Insulin" and the subsections other hypoglycaemic drugs.

·        Table 1 should be editable, not an image. Also, by slightly changing the width of some columns with little content, the font size could be increased slightly, which would be appreciated. Also, in the 'hypoglycaemia risk' column, there are only two categories, low and high risk. If this categorisation is derived from a guideline, the authors should leave it as it is. However, if it is a classification made by the authors, I think there could be a category of "medium risk". For example, in the manuscript it says:  “Hypogly-425 cemia remains a major adverse effect of meglitinides but generally less so than sulfonylu-426 reas.”. In the table, however, the risk is similar.

·        Line 412: " glycemic agents". I have to assume that these are hypoglycaemic agents.

Author Response

[Comment 1]- First, the paragraphs are too long. Paragraphs should generally be no longer than 12-15 lines. In addition, each paragraph should focus on one topic (or several closely related topics). In other words, the manuscript should be restructured by improving the structure, length, and coherence of the paragraphs.

[Response 1] - we have revised the length of paragraphs to try and decrease length but also maintain focus on topic

[Comment 2] - Second, Table 1 should be placed at the beginning of Section 2 of the manuscript, not after the conclusions. It is the guideline proposed by the authors and should be mentioned at the beginning of Section 2 so that readers can quickly visualize the information in the entire manuscript.

[Response 2] - Table 1's location has been moved to the beginning of section 2

[Comment 3] - Section 2 "Insulin", Section 2.1 "Metformin", etc. This structure is confusing. It implies that metformin is a type of insulin. I have to assume that section 2 is about the most commonly used hypoglycaemic drugs. If so, you can call it "Major hypoglycaemic drugs". If the authors mean something else, call it something else. But don't call section 2 "Insulin" and the subsections other hypoglycaemic drugs.

[Response 3] - We have changed the classification system of the medications to be more appropriate pharmacologically

[Comment 4] - Table 1 should be editable, not an image. Also, by slightly changing the width of some columns with little content, the font size could be increased slightly, which would be appreciated. Also, in the 'hypoglycaemia risk' column, there are only two categories, low and high risk. If this categorisation is derived from a guideline, the authors should leave it as it is. However, if it is a classification made by the authors, I think there could be a category of "medium risk". For example, in the manuscript it says:  “Hypogly-425 cemia remains a major adverse effect of meglitinides but generally less so than sulfonylu-426 reas.”. In the table, however, the risk is similar.

[Response 4] - We have made Table 1 editable, and added a "Moderate Risk" category for meglitinides. We also have increased the font size and changed the table column width

[Comment 5] - Line 412: " glycemic agents". I have to assume that these are hypoglycaemic agents.

[Response 5] - classification of these agents has been standardized throughout as "anti-hyperglycemic agents"

Reviewer 2 Report

Comments and Suggestions for Authors

The review article titled (Hypoglycemic medication management in the perioperative setting: A review and illustrative case) by Goron et al. reviewed the current hypoglycemic medications utilized in perioperative setting and mentioned a recent case underscores the potential adverse effects of GLP-1RA in surgical patients, prompting a discussion on prevention and management strategies. This is a mini review for researchers and graduate students, but unforutnately doe snot have a clear aim (authors move from reviewwing drugs to a case reprot and then move again to the perioperative use of drugs), what was the aim of the review?

I have the following recommendations for improving the article:

1- Title: did not mention the importance of the case report!! whether it was for adverse effects or efficacy of certain drug? this is important and really not clear in the article

2- The title should describe clearly what this case report discussing, the current title is not well formulated and not informative

3- Line 384: authors wrote (miscellaneous drugs), it is my first time to see this description for these class, please refer to a recent pharmacology book to classify the drugs adequately

4- Please separate sulfonylurea from meglitinides

5- The topic is not adequately presneted or related. I think it is better to write a review article or publish a case report. BUt the current format is unrelated and does not have a certain focus or aim

especially authors did not review adequately the adverse effects  of the drugs

6- Then Table 1. discusses the Perioperative considerations for the various hypoglycemic agents.

7-I think the term (hypoglycemic drugs) is not correct and authors must revise the use of this term allover the manuscript

8- Keywords: are not perfect for indexing purposes for this article

9-Ensure every abbreviation is explained at the first appearnace in abstract & then in the body text

10- Authors need to explroe this to better describe the rationale and novelty of the study.

11- WHat was the methodology used in this review article? what inclusion criteria and exclusion criteria?

12- Just 1 table was designed to compare drugs and demonstrate the reviewed data

13-

Comments on the Quality of English Language

moderate revision is required

Author Response

[Comments 1 +2 ] - Title: did not mention the importance of the case report!! whether it was for adverse effects or efficacy of certain drug? this is important and really not clear in the article; The title should describe clearly what this case report discussing, the current title is not well formulated and not informative

[Response 1+2] - The title has been altered to reflect the significance of the case report

[Comment 3]- 3- Line 384: authors wrote (miscellaneous drugs), it is my first time to see this description for these class, please refer to a recent pharmacology book to classify the drugs adequately

[Response 3]- The drugs have been re-classified on a pharmacologic basis

[Comment 4]- Please separate sulfonylurea from meglitinides

[Response 4] - We have kept these together with our new medication classification system since they are both insulin secretagogues, however if desired we can separate

[Comment 5] - The topic is not adequately presented or related. I think it is better to write a review article or publish a case report. But the current format is unrelated and does not have a certain focus or aim especially authors did not review adequately the adverse effects of the agents

[Response 5]- we have altered our article to include a more brief recap of the case report, while also reviewing the adverse effects of all of the agents

[Comment 6] -Then Table 1. discusses the Perioperative considerations for the various hypoglycemic agents.

[Response 6] - The peri-operative considerations for the various agents is now addressed in each section as well as Table 1

[Comment 7] - 7-I think the term (hypoglycemic drugs) is not correct and authors must revise the use of this term allover the manuscript

[Response 7] - The term hypoglycmic drug or agent has been replaced with anti-hyperglyecmic agent

[Comment 8]- Keywords: are not perfect for indexing purposes for this article

[Response 8]- We have altered the keywords to be more appropriate for indexing purposes

[Comment 9] - Ensure every abbreviation is explained at the first appearance in abstract & then in the body text

[Response 9] - No abbreviations are present in the abstract. We have reviewed the body's text and ensured that each abbreviation is explained at first appearance.

[Comment 10] - 10- Authors need to explore this to better describe the rationale and novelty of the study.

[Response 10]- The authors have described briefly in the reworked case report portion the novelty of GLP-1 as a possible contributor to peri-operative dysmotility and complications.

[Comment 11]- What was the methodology used in this review article? what inclusion criteria and exclusion criteria?

[Response 11]- The authors aimed to review the available data on anti-hyperglycemic agents using studies that have been published for the majority within the last 20 years, however there are several older studies cited when providing data regarding physiology. Our review did not exclude based on particular type of study. there were no specific inclusion or exclusion criteria as to demographics, type of study or study-specific variables

[Comment 12]- Just 1 table was designed to compare drugs and demonstrate the reviewed data

[Response 12]- The authors felt that one table would be sufficient to highlight the key points of all agents

Round 2

Reviewer 2 Report

Comments and Suggestions for Authors

The revised version of the review article  titled (Hypoglycemic medication management in the perioperative setting: A review and illustrative case) by Gonor et al. was effeciently improved compared to the original version.
I think it is now will be useful for researchers in the field of diabetes medications 
Thanks for the authors for the review 

Comments on the Quality of English Language

Fine